# Identification of CT Imaging Phenotypes of Colorectal Liver Metastases from Radiomics Signatures—Towards Assessment of Interlesional Tumor Heterogeneity

**DOI:** 10.3390/cancers14071646

**Published:** 2022-03-24

**Authors:** Hishan Tharmaseelan, Alexander Hertel, Fabian Tollens, Johann Rink, Piotr Woźnicki, Verena Haselmann, Isabelle Ayx, Dominik Nörenberg, Stefan O. Schoenberg, Matthias F. Froelich

**Affiliations:** 1Department of Radiology and Nuclear Medicine, University Medical Center Mannheim, Medical Faculty Mannheim of the University of Heidelberg, 68167 Mannheim, Germany; hishan.tharmaseelan@medma.uni-heidelberg.de (H.T.); alexander.hertel@umm.de (A.H.); fabian.tollens@medma.uni-heidelberg.de (F.T.); johann.rink@medma.uni-heidelberg.de (J.R.); piotrekwoznicki@gmail.com (P.W.); isabelle.ayx@umm.de (I.A.); dominik.noerenberg@umm.de (D.N.); stefan.schoenberg@umm.de (S.O.S.); 2Institute of Clinical Chemistry, University Medical Center Mannheim, Medical Faculty Mannheim of the University of Heidelberg, 68167 Mannheim, Germany; verena.haselmann@medma.uni-heidelberg.de

**Keywords:** radiomics, computed tomography, colorectal cancer, metastasis, liver metastases

## Abstract

**Simple Summary:**

Tumoral heterogeneity, which is a major challenge in therapy planning, is often characterized by genetic alterations. In this study, an image-based approach was used to identify metastases subtypes by radiomics features. Feature selection and reduction using Pearson correlation threshold and LASSO regression resulted in four final features. After unsupervised clustering and following visual assessment, five lesion clusters could be identified and defined, which had a significant (*p* < 0.01) correlation with sex, primary location, T- and N-status, and mutational status.

**Abstract:**

(1) Background: Tumoral heterogeneity (TH) is a major challenge in the treatment of metastatic colorectal cancer (mCRC) and is associated with inferior response. Therefore, the identification of TH would be beneficial for treatment planning. TH can be assessed by identifying genetic alterations. In this work, a radiomics-based approach for assessment of TH in colorectal liver metastases (CRLM) in CT scans is demonstrated. (2) Methods: In this retrospective study, CRLM of mCRC were segmented and radiomics features extracted using pyradiomics. Unsupervised k-means clustering was applied to features and lesions. Feature redundancy was evaluated by principal component analysis and reduced by Pearson correlation coefficient cutoff. Feature selection was conducted by LASSO regression and visual analysis of the clusters by radiologists. (3) Results: A total of 47 patients’ (36% female, median age 64) CTs with 261 lesions were included. Five clusters were identified, and the categories small disseminated (*n* = 31), heterogeneous (*n* = 105), homogeneous (*n* = 64), mixed (*n* = 59), and very large type (*n* = 2) were assigned based on visual characteristics. Further statistical analysis showed correlation (*p* < 0.01) of clusters with sex, primary location, T- and N-status, and mutational status. Feature reduction and selection resulted in the identification of four features as a final set for cluster definition. (4) Conclusions: Radiomics features can characterize TH in liver metastases of mCRC in CT scans, and may be suitable for a better pretherapeutic classification of liver lesion phenotypes.

## 1. Introduction

Metastatic colorectal cancer (mCRC) is the third most common cancer worldwide [1]. Modern and personalized tumor treatments have substantially increased the survival rates of colorectal cancer (CRC) patients in recent decades. Yet, in a metastatic stage, CRC has a five-year survival rate of just 14.7%, compared to 90.6% in the localized stage [2]. At the time of diagnosis, 22% of patients present in the metastatic stage, i.e., showing synchronous metastases. Of the remaining patients in the localized or regional stage, a relevant proportion will develop metastases in the course of the disease.

Treatment in the metastatic stage is based on chemotherapy, targeted treatment, i.e., antibody treatment with anti-Epidermal Growth Factor Receptor (EGFR) or anti-Vascular Endothelial Growth Factor (VEGF) antibodies, interventional radiology therapy, and surgery [3]. As cancer therapy becomes more and more personalized, the precise determination of patient characteristics is increasingly important. For therapy planning, imaging—especially computed tomography (CT)—currently plays a pivotal role for primary staging [3], while alternative approaches like PET/MRI are emerging [4].

In the context of cancer treatment, tumoral heterogeneity (TH) is a well-recognized challenge, and therefore, focus of research, particularly in molecular biology, as it may be associated with the poorer response of certain lesions. TH has been described in a variety of tumors and can manifest itself in many ways, such as genetic, metabolic, or epigenetic alterations [5]. For instance, TH-related therapy-relevant mutations in colorectal carcinoma have been identified in individual lesions of patients, while other lesions in the same patient remain unaffected from mutations [6].

During tumor development, tumoral biology can change due to mechanisms of tumor evolution, but it can also be influenced by therapy-induced tumoral escape selection pressure and other factors. In the context of TH, a distinction must be made between inter- and intra- lesional heterogeneity. Intralesional heterogeneity describes the possible molecular variations within a coherent lesion, while interlesional heterogeneity describes the differences between multiple lesions within one patient [5]. While locoregional interventional therapy is gaining in importance, interlesional heterogeneity, among other factors, may pose a challenge, as biopsies can typically only be obtained from singular lesions, and may not be representative of the overall tumoral biology in a patient. This finding is supported by comprehensive sequencing approaches of primary tumor and metastatic lesions, which have unraveled a high degree of genetic heterogeneity in colorectal cancer occurring during tumor evolution, which can be an obstacle for optimizing treatment choice [7]. Therefore, liquid profiling (LP) is becoming more widely used in clinical routine, as it can detect emerging resistance-mediating genetic alterations during therapy. These variations are usually subclonal, representing intertumoral heterogeneity. However, in oligometastatic disease, the identification of the corresponding resistance-driving lesions remains challenging. Regarding the aforementioned problems from a clinical standpoint, assessments of TH from imaging based on radiomics and clustering would add valuable clinical information and help to stratify treatment options more precisely.

In recent years, quantitative imaging biomarkers (QIB) and quantitative image analysis, namely radiomics, have emerged and are deemed very promising in this field [8]. Radiomics is a method for the extraction of vast amounts of quantitative, mostly nonhuman readable image features and texture analysis from radiologic imaging [9].

First analyses showed the potential of radiomics and lesion distribution patterns for outcome assessment in metastatic colorectal cancer (mCRC) [10] and other tumoral entities [11,12]. Tumor heterogeneity evaluations on computed tomography scans of upper tract urothelial carcinoma have been shown to be able to differentiate between muscle-invasive and nonmuscle invasive tumors [13]. Also, radiomics features have been linked to mutational patterns. In particular, the presence of KRAS, NRAS, and BRAF status in liver metastases has been associated with radiomics and semantic imaging features [14] as well as the corresponding response to targeted therapy [15].

Yet, a lesion-wise, image-based cluster analysis with a clear focus on imaging-based interlesional heterogeneity has not been performed on liver metastases of colorectal cancer. Therefore, this study aimed to identify radiomics-based lesion phenotypes by unsupervised clustering, which might serve as a basis for the assessment of interlesional heterogeneity.

## 2. Materials and Methods

### 2.1. Patient Collective and Imaging Protocols

For this retrospective, single-center, IRB-approved study, 47 patients with proven colorectal cancer and colorectal liver metastases (CRLM) were identified. All subjects were scanned in a commonly used CT scanner for in- and outpatient oncologic cases in Mannheim University Medical Center, i.e., a 16-slice CT scanner (Siemens SOMATOM Emotion^®^, Siemens Healthcare GmbH, 91052 Erlangen, Germany), from 2012 to 2019. To maintain the comparability of the radiomics features between individual patients, only portal venous phase contrast-enhanced axial CT datasets with a slice thickness of 1.5 mm which had a histologically confirmed colorectal adenocarcinoma were included.

In accordance with German medical data privacy guidelines, image DICOM data were anonymized via Aycan-Workstation (Aycan Digitalsysteme GmbH, 97076 Wuerzburg, Germany). CT images were obtained with a tube voltage of 130 kV and tube current modulation. Reconstruction was performed on the B30s kernel. Clinical data, including TNM-status, microsatellite instability, KRAS/NRAS/BRAF mutational status, and if surgical or chemotherapeutic treatment was carried out before the date of CT, were retrieved from the PACS system. For cases in which a retrospective determination of the TNM status from the documentation was not possible, an image-based staging was performed by a clinical radiologist (M.F.F. with > 4 years of experience in oncologic imaging and mCRC diagnostics).

### 2.2. Image Analysis and Segmentation

For each patient, up to 28 liver lesions were segmented into 1.5 mm axial slices using a semi-automated approach in 3D Slicer (version 4.11). Segmentations were performed by a medical student (H.T., two years of experience in segmentation) and were reviewed by a clinical radiologist (M.F.F.).

### 2.3. Radiomics Feature Extraction

After the segmentation, features for the defined regions of interest (ROI) were extracted in the open-source radiomics platform, Pyradiomics (version 3.0.1), in python. The extraction parameters and settings used in this analysis can be found in the Appendix A (Appendix A). Radiomics features from the following categories were extracted for each ROI: “firstorder”, describing the distribution of hounsfield intensities without comparing to the spatial reference; “shape”, describing the 3D and 2D shape of the ROI; “glcm” [Gray Level Cooccurrence Matrix], “gldm” [Gray Level Dependence Matrix], “glrlm” [Gray Level Run Length Matrix], “glszm” [Gray Level Size Zone Matrix], and “ngtdm” [Neighboring Gray Tone Difference Matrix], which are textural features that describe the distribution of gray tones calculated by statistical comparison with the surrounding voxels.

Following per-lesion feature extraction, the results were exported and prepared for further analysis in dedicated statistics and data analytics software.

### 2.4. Clustering, Feature Selection, and Statistical Analysis

The statistical analysis was performed in R [11] and RStudio (version 1.3.1093, Boston, MA, USA). The list of utilized software packages can be found in Appendix A. All demographic and clinical parameters were summarized with median and interquartile range (IQR).

Feature normalization was performed using the z-score (Equation (1)), where each feature value X was scaled with the mean (µ) and standard deviation (σ) of the lesion features.
z = ((X − µ))/σ(1)

After normalization, the feature-to-feature correlation was calculated using the Pearson correlation coefficient (PCC). Subsequently, normalized and correlated features were visualized in an unclustered heatmap. Following that, 100-times repeated k-means clustering of lesions by patient and radiomic features was applied to differentiate potentially clinically relevant cohorts and displayed in an additional heatmap.

PCA was performed to investigate a possible redundancy of features by determining the explained variance of each dimension. To reduce the number of redundant features a pairwise correlation filter was applied and omitted those above a threshold PCC of 0.75. The corresponding, nonredundant features were analyzed by least absolute shrinkage and selection operator (LASSO) regression to identify and rank the most relevant features for the differentiation between cluster groups. Reduced heatmaps were created for the final feature set.

### 2.5. Cluster Analysis

Groups resulting from the k-means clustering were reviewed both visually and quantitatively based on the reduced feature sets. Clinically meaningful descriptors were defined both based on lesion size, feature texture quantification, and visual impression by two experienced radiologists. The collected clinical parameters were correlated with the clusters by applying a Chi-squared test.

## 3. Results

### 3.1. Patient Collective

Based on the inclusion criteria, a total of 316 CT scans of patients with suspected colorectal cancer were identified. Two hundred and sixty-six patients were excluded because they did not show liver metastases at the time of diagnosis. For the remaining 50 patients, 3 were excluded due to histological statuses which were not consistent with colorectal adenocarcinoma (Figure 1).

As a result, a total of 47 patients were enrolled in this study. In the reported study population, 36% of the patients were female and had a median age of 64. At the time of the CT scan, 14 patients had previously undergone surgery and 25 had undergone systemic treatment. Patient characteristics are summarized in Table 1. For all patients, segmentation of liver metastases was performed according to the approach presented in Materials and Methods. An example of the segmentations for one patient is shown in Figure 2.

### 3.2. Cluster Analysis

After feature extraction and standardization, an unclustered heatmap of all lesions was created (Figure 3a). Unsupervised clustering of lesions in five, and features in seven groups was performed with the integrated k-means clustering method of the “ComplexHeatmap”-package in R. The total number of lesions was 31, 105, 64, 59, and 2 for each respective cluster (Table 2). Visualization of the clusters in a heatmap (Figure 3b) revealed a high redundancy within a significant amount of radiomics features. The feature redundancy was further analyzed by applied PCA.

### 3.3. Reduction of Feature Redundancy

PCA identified a significant number of redundant features within only a few dimensions (). Consequently, feature redundancy reduction with a PCC threshold of 0.75 was performed. As a result of these steps, the number of features was reduced from 65 to 14, as shown in Figure 4.

### 3.4. Feature Importance Assessment

To assess and quantify feature importance, LASSO regression was performed. The application of regression resulted in a further feature reduction from 14 to 4 features (Appendix A). This led to the identification of “original firstorder Range”, “original gldm DependenceVariance”, “original glrlm RunLengthNonUniformity”, “original glrlm ShortRunLowGrayLevelEmphasis’’ as the final feature set for cluster definition. The features are shown as a heatmap in Figure 5 and a boxplot in Figure 6. Lesion voxel volume was added manually for reference.

### 3.5. Visual Cluster Analysis

Lesions within these five clusters were analyzed visually by two experienced radiologists (M.F.F. and D.N.), and the corresponding categories were assigned: (i) small disseminated, (ii) heterogeneous type, (iii) homogeneous type, (iv) mixed type, and (v) very large type (Figure 7a). An example patient with a relevant degree of imaging-based interlesional heterogeneity is shown in Figure 7b, with a visualization of the first-order radiomics feature range.

### 3.6. Association of Clusters with Clinical Patterns and Parameters

The correlation of clinical parameters with the five previously defined clusters was investigated by applying a Chi-squared test. Male sex was associated with a higher proportion of lesions from cluster 2 (heterogeneous type), while female sex was associated with the presence of cluster 3 (homogeneous type) lesions. Mixed type lesions were associated with higher T-stage, while higher N-stage was associated with small disseminated lesion type. Also, the test showed a significant correlation of lesion type with patient sex (*p <* 0.001), primary tumor location (*p <* 0.001) and mutational status (*p <* 0.001). The results of the associations are shown in Table 2.

## 4. Discussion

In summary, this work demonstrates the feasibility of using radiomics features to identify imaging-based patterns of colorectal liver metastases in an unsupervised approach. It resulted in the proposal of five distinct imaging patterns for mCRC. The presented work applies a strictly imaging-based approach to assess interlesional imaging-heterogeneity based on CT radiomics features, and provides a proof-of-concept for the potential distinction of individual mCRC liver lesions on CT. Nevertheless, the underlying biological ground truth is challenging to identify, because in vivo biopsies of every liver lesion are not clinically applicable from a medical and ethical point of view. By feature redundancy reduction, the number of radiomics features needed to identify each subtype could be reduced to four radiomics markers in total. These results may contribute to the development of novel imaging-based biomarkers in the context of metastatic colorectal cancer.

While the association of imaging-based lesion-specific markers with underlying tumoral mutational patterns may be difficult to investigate in clinical routine, it has already been addressed in autopsy studies. Results published by Siravegna et al. [16] demonstrated a high degree of interlesional tumoral heterogeneity in colorectal liver metastases with corresponding mutational patterns, evolutionary mechanisms and correlating response. These mechanisms may underlie the CT-based interlesional heterogeneity assessment presented in this work. While biopsy-based mutational analyses of every lesion may not be clinically feasible, future application of Liquid Profiling may help to close this gap.

In contrast to the proposed imaging-focused phenotype approach, other groups have investigated the association of radiomics patterns with overall tumoral mutational patterns, partly not taking tumoral heterogeneity into account. Aerts et al., investigated the prognostic value of tumor phenotypes defined by radiomics for both lung cancer and head-and-neck cancer [17]. Lafata et al., analyzed the association of radiomics features with underlying tumor mutational and cfDNA patterns [18] and performed first associations with p53 status, however, only focused on primary lung tumors and not on multiple metastases. A recently published study by Starmans et al., demonstrated that radiomics and machine learning features from CT can predict histopathological tumor growth patterns in colorectal liver metastases [19]. Further analyses have been able to demonstrate an association of radiomics parameters and mutational and histopathological patterns [20,21], often referred to as radiogenomics [22]. In the context of the literature, this work with an unsupervised cluster identification focuses on the potential imaging-based assessment of heterogeneity, while many previous works associate radiomics features with clinical and/or mutational information such as radiomics and distribution-based survival prediction in mCRC patients [10,23,24].

Given that this work relied heavily on radiomics features for lesion clustering, the methodology of radiomics and its potential pitfalls are of importance for the clinical applicability and generalizability of the presented results. Routinely collected radiomics features have a high degree of collinearity, which was addressed by feature reduction steps in our methodology. More importantly, the individual values of radiomics features can be influenced by several parameters that were addressed in this study: First, the choice of CT scanner hardware influences the radiomics features [25]. As a result, only imaging data from one CT scanner was included in the study. Second, the size of the region of interest used influences feature stability [26]. In this study, no arbitrary ROIs were used, but only individual lesions were segmented. Third, extraction software/feature platform and processing steps after extraction can affect the result [27,28]. To address this issue, the image biomarker standardization initiative definition-based [29] Python package Pyradiomics, and only images from one scanner with a slice thickness of 1.5 mm and B30s kernel reconstruction, were used. Also, coffee-break same day test–retest studies show the variation of radiomics features [30,31]. Yet, repeated measurements of patients on the same day would not be acceptable from an ethical point of view, and therefore, cannot be performed in a patient study.

Radiomics parameters are a very promising target for unsupervised cluster analyses and the identification of lesion subtypes from a methodological point of view. Therefore, radiomics features were employed in this study. The radiomics methodology is very powerful and was, for example, even able to predict the microvascular invasion in hepatocellular carcinoma [32].

The results presented in this work must be interpreted in the context of certain limitations: As stated above, this single-center study only included CT scans from one scanner type. An additional multicenter validation with different scanners might be beneficial. Moreover, there are inherent limitations with radiomics methodologies in terms of feature reproducibility and stability [32]. These may be overcome by the implementation of emerging supervised techniques such as deep learning to increase feature stability.

A widely proposed solution to further increase the stability of quantitative imaging approaches is the application of deep learning reconstruction methods to imaging data [33]. While this approach has shown promising results in the prediction of known phenotypes [34], it has some limitations in an unsupervised approach, because a ground truth for training cannot be defined easily. Especially in this study, deep learning approaches were not feasible, as the goal of the study was to perform unsupervised clustering to identify metastases phenotypes in imaging. Furthermore, there are some limitations with the patient collective due to the retrospective approach of the study: Imaging data were available only with B30s kernel reconstructions and 1.5 mm slice thickness. In addition, some clinical parameters, most importantly information on mutational status, were missing (Table 1) and could not be retrieved retrospectively. In the case of the patients who had already undergone systemic treatment, the specific therapy regimen or information about prior surgical therapy was also not available for all patients. Another limitation is that the appearance of metastases may have been affected by systemic treatment in a subcohort of the reported patient collective. Also, the alterations of radiomics-based lesional patterns during the time course of therapy would be an interesting topic not assessed in this study, which relied only on a singular time point for each patient due to lack of data comparability and availability. While the information on clinical features may be partly inconclusive in this study, a focus was set on comparable imaging data, both in terms of scanner, scanning parameters, and reconstruction algorithms, to avoid image-associated bias to the unsupervised k- means clustering approach.

In summary, this work is a proof of concept for the assessment of intertumoral lesion-specific heterogeneity to identify the tumor phenotype via CT imaging. The implementation of new technologies like photon-counting CT [35] into the clinical routine may help overcome the challenges of feature stability and comparability. Further, the inclusion of clinical and laboratory parameters toward integrated diagnostics, as well as follow-up imaging within the course of systemic treatment, will improve the validity of quantitative image analyses in future studies. Combined analysis of genetic information obtained by LP and radiogenomic feature extraction could help deepen our understanding of emerging resistance mechanisms, topologically assign subclonal variations, and thus, precisely adapt individual therapeutic approaches to real-time tumor evolution.

## 5. Conclusions

Radiomics features can characterize TH in liver metastases of mCRC in CT scans, and may be suitable to improve pretherapeutic classifications of liver lesion phenotypes.

## Figures and Tables

**Figure 1 cancers-14-01646-f001:**
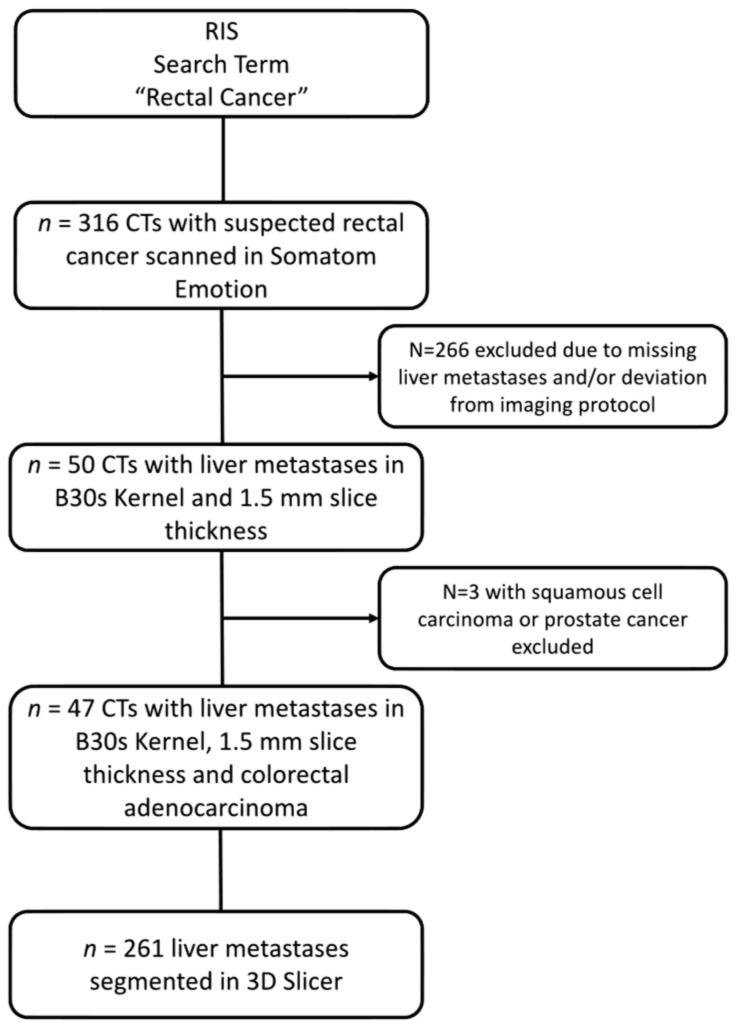
Patient collective definition consort flow diagram.

**Figure 2 cancers-14-01646-f002:**
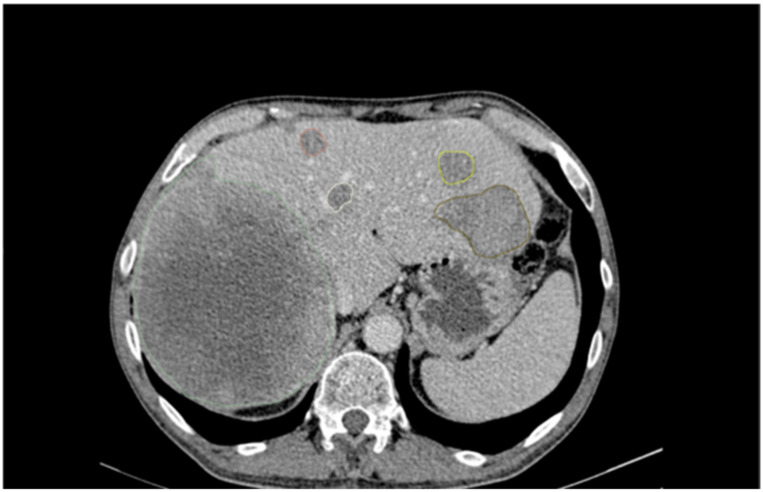
Example segmentations of a patient.

**Figure 3 cancers-14-01646-f003:**
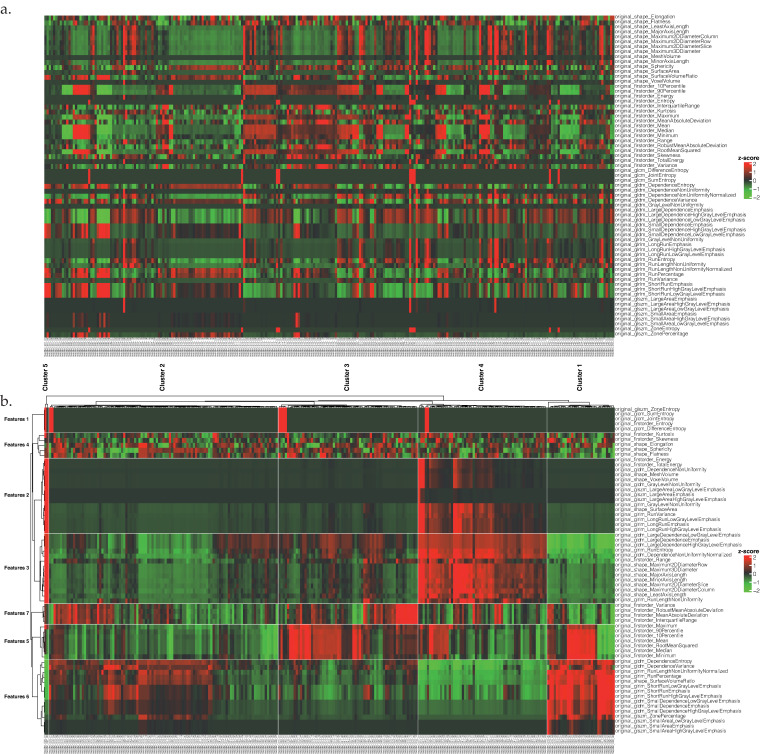
(**a**) Radiomics feature information of all lesions without clustering. (**b**) Unsupervised clustering of lesions and features.

**Figure 4 cancers-14-01646-f004:**
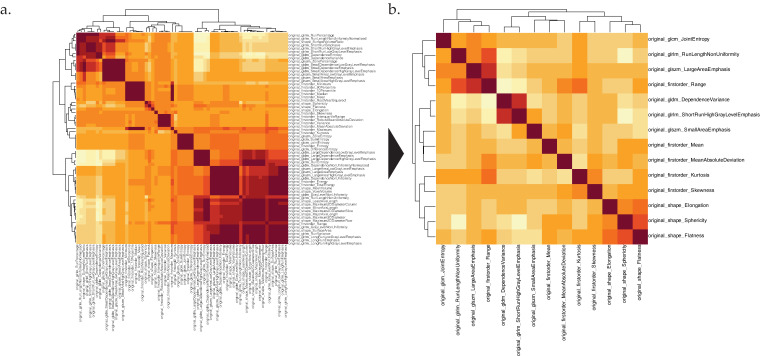
(**a**) Pearson correlation coefficient heatmap with all features. (**b**): Heatmap with redundancy reduced features after correlation threshold of 0.75.

**Figure 5 cancers-14-01646-f005:**
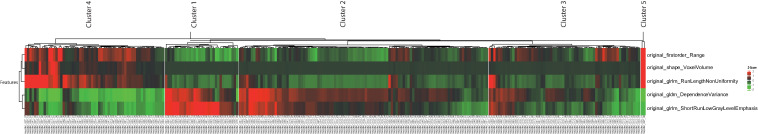
Feature reduced heatmap clustered by lesions and features (voxel volume was added for reference).

**Figure 6 cancers-14-01646-f006:**
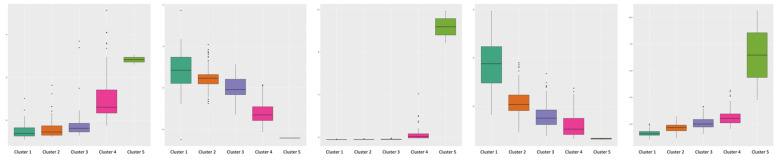
Boxplot diagrams for the final feature set (voxel volume was added manually).

**Figure 7 cancers-14-01646-f007:**
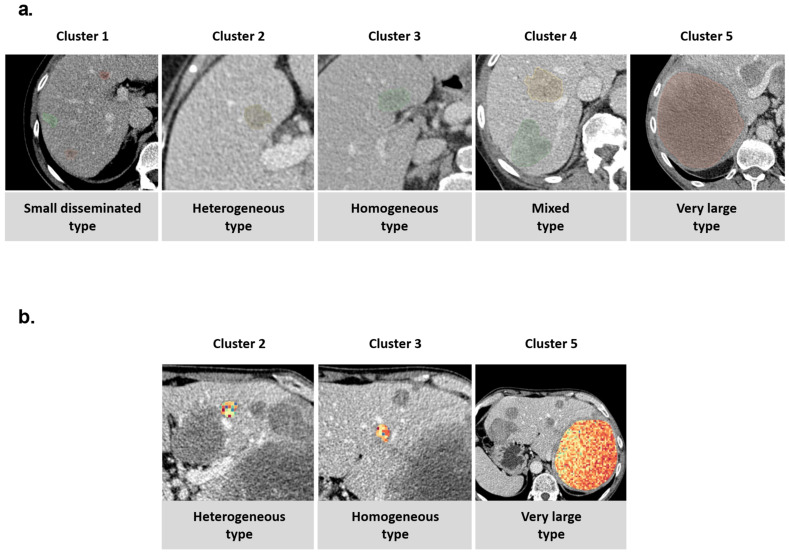
(**a**) Visually defined clinical groups of CRC liver metastases. (**b**) Patient with relevant interlesional heterogeneity and presence of lesions from multiple clusters. Visualization of feature firstorder_Range also shows a relevant degree of intralesional heterogeneity of the feature.

**Table 1 cancers-14-01646-t001:** Patient characteristics. Median and IQR.

Variable		Overall
*n*		47
Age at CT (median [IQR])		65.79 [56.99, 74.62]
Sex (%)		
	F	17 (36.2%)
	M	30 (63.8%)
Tumor Location (%)		
	Colon	1 (2.1%)
	Colon asc	2 (4.3%)
	Colon desc	3 (6.4%)
	Colon tran.	3 (6.4%)
	Rectum	29 (61.7%)
	Rectosigmoid Junction	2 (4.3%)
	Sigma	7 (14.9%)
T-Stage (%)		
	T1	2 (4.3%)
	T2	4 (8.5%)
	T3	24 (51.1%)
	T4	15 (31.9%)
	Tx	2 (4.3%)
N-Stage (%)		
	N0	8 (17.0%)
	N1	18 (38.3%)
	N2	20 (42.6%)
	Nx	1 (2.1%)
M-Stage (%)		
	M1	47 (100.0%)
pre-CT Surgery (%)		
	No	6 (30.0%)
	Yes	14 (70.0%)
	Unknown	27
pre-CT Chemotherapy (%)		
	No	21 (46.7%)
	Yes	24 (53.3%)
	Unknown	2
KRAS-Mutation (%)		
	No	23 (67.6%)
	Yes	11 (32.4%)
	Unknown	13
NRAS-Mutation (%)		
	No	32 (94.1%)
	Yes	2 (5.9%)
	Unknown	13
BRAF-Mutation (%)		
	No	13 (86.7%)
	Yes	2 (13.3%)
	Unknown	32
MSS/MSI (%)		
	MSI	1 (5.0%)
	MSS	19 (95.0%)
	Unknown	27

**Table 2 cancers-14-01646-t002:** Clusters and per-lesion patient characteristics.

Variable		Cluster 1	Cluster 2	Cluster 3	Cluster 4	Cluster 5	*p*
*n* (lesions)		31	105	64	59	2	
Sex (%)	F	18 (20.22%)	14 (15.73%)	34 (38.2%)	22 (24.72%)	1 (1.12%)	<0.001
	M	13 (7.56%)	91 (52.91%)	30 (17.44%)	37 (21.51%)	1 (0.58%)	
Tumor Location (%)	Colon	0 (0%)	0 (0%)	2 (100%)	0 (0%)	0 (0%)	<0.001
	Colon asc.	0 (0%)	8 (50%)	4 (25%)	4 (25%)	0 (0%)	
	Colon desc.	0 (0%)	8 (38.1%)	3 (14.29%)	10 (47.62%)	0 (0%)	
	Colon tran.	3 (21.43%)	1 (7.14%)	2 (14.29%)	8 (57.14%)	0 (0%)	
	Rectum	25 (15.15%)	74 (44.85%)	44 (26.67%)	21 (12.73%)	1 (0.61%)	
	Rectosigmoid Junction	0 (0%)	0 (0%)	4 (44.44%)	5 (55.56%)	0 (0%)	
	*Sigma*	3 (8.82%)	14 (41.18%)	5 (14.71%)	11 (32.35%)	1 (2.94%)	
T-Stage (%)	T1	0 (0.0%)	2 (1.9%)	3 (4.7%)	1 (1.7%)	0 (0.0%)	0.009
	T2	6 (19.4%)	13 (12.4%)	3 (4.7%)	6 (10.2%)	0 (0.0%)	
	T3	16 (51.6%)	57 (54.3%)	22 (34.4%)	24 (40.7%)	1 (50.0%)	
	T4	6 (19.4%)	33 (31.4%)	32 (50.0%)	28 (47.5%)	1 (50.0%)	
	Tx	3 (9.7%)	0 (0.0%)	4 (6.2%)	0 (0.0%)	0 (0.0%)	
N-Stage (%)	N0	2 (6.5%)	9 (8.6%)	12 (18.8%)	5 (8.5%)	0 (0.0%)	<0.001
	N1	9 (29.0%)	68 (64.8%)	14 (21.9%)	34 (57.6%)	2 (100.0%)	
	N2	17 (54.8%)	28 (26.7%)	36 (56.2%)	20 (33.9%)	0 (0.0%)	
	Nx	3 (9.7%)	0 (0.0%)	2 (3.1%)	0 (0.0%)	0 (0.0%)	
pre-CT Surgery (%)	No	7 (28%)	13 (52%)	3 (12%)	2 (8%)	0 (0%)	NA
	Yes	13 (12.26%)	43 (40.57%)	36 (33.96%)	14 (13.21%)	0 (0%)	
pre-CT Chemotherapy (%)	No	11 (9.48%)	61 (52.59%)	23 (19.83%)	20 (17.24%)	1 (0.86%)	0.006
	Yes	20 (15.27%)	38 (29.01%)	41 (31.3%)	31 (23.66%)	1 (0.76%)	
KRAS-Mutation (%)	No	21 (16.15%)	37 (28.46%)	37 (28.46%)	35 (26.92%)	0 (0%)	<0.001
	Yes	2 (2.3%)	51 (58.62%)	20 (22.99%)	13 (14.94%)	1 (1.15%)	
NRAS-Mutation (%)	No	16 (7.77%)	87 (42.23%)	54 (26.21%)	48 (23.3%)	1 (0.49%)	<0.001
	Yes	7 (63.64%)	1 (9.09%)	3 (27.27%)	0 (0%)	0 (0%)	
BRAF-Mutation (%)	No	7 (7.53%)	26 (27.96%)	33 (35.48%)	26 (27.96%)	1 (1.08%)	<0.001
	Yes	2 (5.88%)	29 (85.29%)	3 (8.82%)	0 (0%)	0 (0%)	
MSS/MSI (%)	MSI	0 (0%)	0 (0%)	0 (0%)	2 (100%)	0 (0%)	0.095
	MSS	10 (6.9%)	63 (43.45%)	43 (29.66%)	28 (19.31%)	1 (0.69%)	

## Data Availability

The data presented in this study are available on request from the corresponding author. The data are not publicly available due to German EHR privacy guidelines.

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
