# Peer review of "Identification of CT Imaging Phenotypes of Colorectal Liver Metastases from Radiomics Signatures—Towards Assessment of Interlesional Tumor Heterogeneity"

_cancers, 2022, doi:10.3390/cancers14071646_

Round 1
Reviewer 1 Report
Authors investigated the identification of CT imaging phenotypes of liver metastasis of colorectal cancers using "Radiomics" signatures. The aim of this study is assess the intralesional tumor heterogeneity, I feel Radiomics is a pretty new research field and trials such as the study shown in this manuscript will be necessary to develop this new technology.
The following things should be modified or answered by authors. Regarding table 2, the numbers of patients divided by clusters should be the same number in all subcategories such as tumor location, T-stage or other, or should be many numbers which omitted the "unknown" numbers shown in Table 1. However, many numbers are not fit.
How does this radiomics-clustering contribute to clinical features such as prognosis, diagnostic advantages or others? I think authors can compare between actual clinical findings and radiomics-clustering. If there is less correlation, some findings from radiomics will support to improve this weakness of choosing therapies by radiomics. This point would be the most important isseu to employ radiomics in clinics.
Author Response
The author's note to reviewer is attached as PDF.

Reviewer 2 Report
Interesting study on the radiomics of CRC mets in liver.
Please delete/avoid personal verbal reference (i.e., “I”, “we”, “our”) in the manuscript. Aim at reaching neutral formal style. Thank you.
Avoid unnecessary wording and terms used in verbal communication (i.e., “then”, “again”, etc). Simple and direct style text throughout the manuscript will improve it and reduce the number of characters.
Specific comments:
Abstract: provide numbers in results as well as significant values.
Introduction: 2 first paragraphs may be summarized.
Materials and methods: please provide a more detailed description of the validation dataset if any.
Results: clear and concise.
Discussion: Interesting but needs to be focused on the results of the study. Avoid verbiage. It would be interesting to have a more thorough discussion of the significant values.
Author Response

(The authors gave the same response as above.)

Reviewer 3 Report
The submitted manuscript by Tharmaseelan and co-workers introduced an interesting aspect. However, two major points highly limit this work. The number of patients included for a very common malignant disease is low (n=47 pts). This questions should be addressed in a multicenter retrospective fashion. Second, the results, in particular figures presented are not sufficient for publication. There is no clear legend available, low quality, unreadable and with 7 too much.
Author Response

(The authors gave the same response as above.)

Round 2
Reviewer 1 Report
Authors adequately answered to reviewer's problems and modified their manuscript. I feel now this manuscript will be suitable for publication in this journal.
Reviewer 2 Report
no additional comments.
Authors addressed the questions.
Reviewer 3 Report
based on the comments of the other reviewers and the revisions made so far by the authors, I agree with the process of acceptance for this article.